# Unsupervised Intuitive Physics from Past Experiences

## Abstract

We consider the problem of learning models of intuitive physics from raw, unlabelled visual input. Differently from prior work, in addition to learning general physical principles, we are also interested in learning "on the fly" physical properties specific to new environments, based on a small number of environment-specific experiences. We do all this in an unsupervised manner, using a meta-learning formulation where the goal is to predict videos containing demonstrations of physical phenomena, such as objects moving and colliding with a complex background. We introduce the idea of summarizing past experiences in a very compact manner, in our case using dynamic images, and show that this can be used to solve the problem well and efficiently. Empirically, we show, via extensive experiments and ablation studies, that our model learns to perform physical predictions that generalize well in time and space, as well as to a variable number of interacting physical objects.

## 1 Introduction

Many animals possess an intuitive understanding of the physical world. They use this understanding to accurately and rapidly predict events from sparse sensory inputs. In addition to general physical principles, many animals also learn specific models of new environments as they experience them over time. For example, they can explore an environment to determine which parts of it can be navigated safely and remember this knowledge for later reuse.

Authors have looked at equipping artificial intelligences (AIs) with analogous capabilities, but focusing mostly on performing predictions from instantaneous observations of an environment, such as a few frames in a video. However, such predictions can be successful only if observations are combined with sufficient prior knowledge about the environment. For example, consider predicting the motion of a bouncing ball. Unless key parameters such as the ball's elasticity are known a priori, it is impossible to predict the ball's trajectory accurately. However, after observing at least one bounce, it is possible to infer some of the parameters and eventually perform much better predictions.

In this paper, we are interested in learning intuitive physics in an entirely unsupervised manner, by passively watching videos. We consider situations in which objects interact with scenarios that can only be *partially* inferred from their appearance, but that also contain objects whose parameters cannot be confidently predicted from appearance alone (fig. 1). Then, we consider learning a system that can observe a few physical experiments to infer such parameters, and use this knowledge to perform better predictions in the future.

Our model has three goals. First, it must learn without the use of any external or ad-hoc supervision. We achieve this by training our model from raw videos, using a video prediction error as a loss.

Second, our model must be able to extract information about a new scenario by observing a few experiments, which we formulate as meta-learning. We also propose a simple representation of the experiments based on the concept of "dynamic image" that allows to process long experiments more efficiently than using a conventional recurrent network.

Third, our model must learn a good representation of physics without access to any explicit or external supervision. Instead, we propose three tests to support this hypothesis. (i) We show that the model can predict far in the future, which is a proxy to *temporal invariance*. (ii) We further show that the model can extend to scenarios that are geometrically much larger than the ones used for training, which is a proxy to *spatial invariance*. (iii) Finally, we show that the model can generalize to several

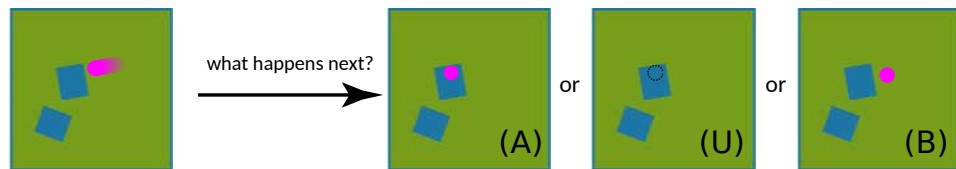

Figure 1: **Probing a new environment.** Given the input showing a (purple) ball moving from right to left, three possible future outcomes are possible: the ball passes above (A), passes under (U), or bounces (B) off the object. This cannot be predicted from appearance alone. Instead, our model learns the property of the object by probing each new environment via a small set of experiments.

moving objects, which is a proxy to *locality*. Locality and time-space invariance are of course three key properties of physical laws and thus we should expect any good intuitive model of physics to possess them.

In order to support these claims, we conduct extensive experiments in simulated scenarios, including testing the ability of the model to cope with non-trivial visual variations of the inputs. While the data is simpler than a real-world application, we nevertheless make substantial progress compared to previous work, as discussed in section 2. We do so by learning from passive, raw video data a good model of dynamics and collisions that generalizes well spatially, temporally, and to a variable number of objects. The scalability of our approach, via the use of the dynamic image, is also unique. Finally, we investigate the problem of learning the parameters of new scenarios on the fly via experiences and we propose an effective solution to do so.

## 2 RELATED WORK

A natural way to represent physics is to manually encode every object parameter and physical property (e.g., mass, velocity, positions) and use supervision to make predictions. This has been widely used to represent and propagate physics (Wu et al., 2015; 2016; Battaglia et al., 2016; Chang et al., 2017; Mrowca et al., 2018; Sanchez-Gonzalez et al., 2018). If models like Wu et al. (2015; 2016) also estimate environment parameters, these works rely on a physics engine that assumes strong priors about the scenario, while our approach does not require such constraint.

Inspired by the recent successes of Convolutional Neural Networks (CNNs (Krizhevsky et al., 2012)) and their application to implicit representation of dynamics (Ondruska & Posner, 2016; Oh et al., 2015; Chiappa et al., 2017; Bhattacharyya et al., 2018), researchers (Watters et al., 2017; Ehrhardt et al., 2019; Kansky et al., 2017) have tried to base their approaches on visual inputs. They learn from several frames of a scene to regress the next physical state of a system. In general these approaches learn an implicit representation of physics (Ehrhardt et al., 2019; Watters et al., 2017) as a tensor state from recurrent deep networks.

Such models are mostly supervised using ground-truth information about key physical parameters (e.g., positions, velocities, density) during training. While these approaches require an expensive annotation of data, other works have tried to learn from unsupervised data as well. Researchers have successfully learned unsupervised models either through active manipulation (Agrawal et al., 2016; Denil et al., 2016; Finn et al., 2016a), using the laws of physics (Stewart & Ermon, 2017), using dynamic clues and invariances (Greff et al., 2017; van Steenkiste et al., 2018) or features extracted from unsupervised methods (Finn et al., 2016b; Ehrhardt et al., 2018). Fragkiadaki et al. (2016) also used an unsupervised system like ours, however, they assumed the rendering system and the number of objects to be known, which we do not, which prevented the internal representation to fail over time. Perhaps most related to our approach is the work of Wang et al. (2018), where the model is learnt using future image prediction on a simple synthetic task and then transferred to real world scenarios. They also demonstrate long-term dynamic predictions, however they did not generalize to different background/number of balls.

In other works, models are taught to answer simple qualitative questions about a physical setup, such as: the stability of stacks of objects (Battaglia et al., 2013; Lerer et al., 2016; Li et al., 2017; Groth et al., 2018), the likelihood of a scenario (Riochet et al., 2018), the forces acting behind a scene (Wu et al., 2017; Mottaghi et al., 2016) or properties of objects through manipulation (Agrawal et al., 2016; Denil et al., 2016). Other papers compromise between qualitative and quantitative predictions and focus on *plausibility* (Tompson et al., 2016; Ladický et al., 2015; Monszpart et al., 2016).

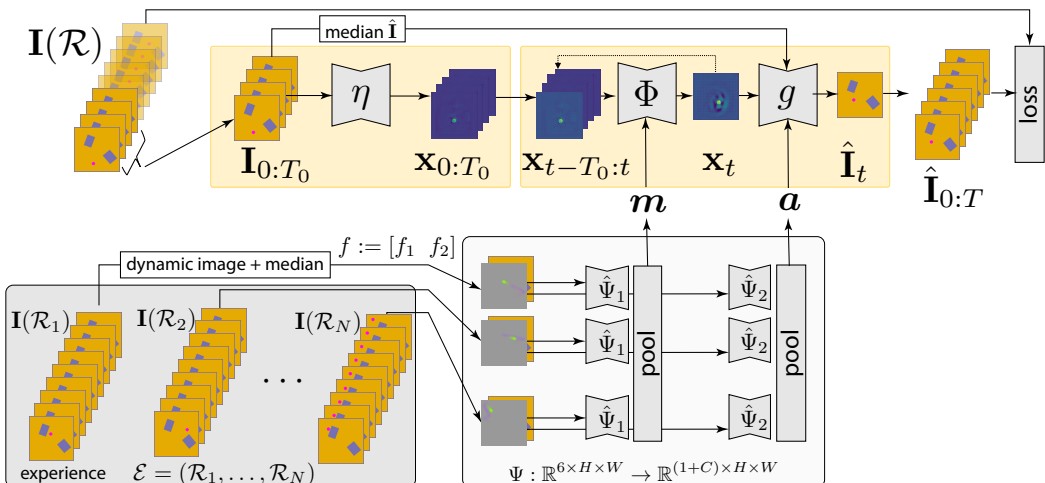

Figure 2: **Overview of our method.** $\Psi$ (bottom right block) acts as a meta-learning module. It takes as input past experiments compressed into dynamic images alongside with the median image and learn to optimise $m$ and $a$ to minimise the final loss. $\eta$ (top-left yellow block) extracts states $\mathbf{x}_{0:T_0}$ which are carried forward with the auto-regressive predictor $\Phi$. Finally $g$ renders frame $\hat{\mathbf{I}}_t$ from $\mathbf{x}_t$.

Finally similar problems can be found in online learning settings (Nagabandi et al., 2019a;b). These works also use meta-learning but, differently from ours, allow their models to learn and adapt at test time from feedbacks.

## 3 METHOD

We describe our model, illustrated in fig. 2, starting from the input and output data.

A *scenario* $S$ is a physical environment that supports moving objects interacting with it. In this paper, we take as scenarios 2.1D environments containing obstacles and we consider rolling balls as moving objects. Hence, interactions are in the form of bounces. Formally, a scenario is defined over a lattice $\Omega = \{0, \dots, H-1\} \times \{0, \dots, W-1\}$ and is specified by a list of obstacles $\mathcal{S} = \{(O_j, b_j), j = 1, \dots, K\}$. Here $O_j \subset \Omega$ is the shape of an obstacle and $b_j \in \{B, A, U\}$ is a flag that tells whether the ball bounces against it (B), passes above it (A), or passes under it (U). Obstacles do not overlap.

A *run* is a tuple $\mathcal{R} = (\mathcal{S}, \mathbf{y})$ associating a scenario $\mathcal{S}$ with a trajectory $\mathbf{y} = (y_t \in \Omega, t = 0, \dots, T-1)$ for the ball (this is trivially extended to multiple balls). Scenarios and runs are sensed visually. The symbol $\mathbf{I}(\mathcal{S}) : \Omega \to \mathbb{R}^{3 \times H \times W}$ denotes the image generated by observing a scenario with no ball and $\mathbf{I}_t(\mathcal{R}) = \mathbf{I}(\mathcal{S}, y_t)$ is the image generated by observing the ball at time $t$ in a run. The symbol $\mathbf{I}(\mathcal{R}) = (\mathbf{I}_t(\mathcal{R}), t = 0, \dots, T-1)$ denotes the collection of all frames in a run, which can be thought of as a video.

We are interested in learning intuitive physics with no explicit supervision on the object trajectories, nor an explicit encoding of the laws of mechanics that govern the motion of the objects and their interactions and collisions with the environment. Hence, we cast this problem as predicting the video $\mathbf{I}(\mathcal{R})$ of the trajectory of the objects given only a few initial frames $\mathbf{I}_{0:T_0}(\mathcal{R}) = (\mathbf{I}_0(\mathcal{R}), \dots, \mathbf{I}_{T_0-1}(\mathcal{R}))$, where $T_0 \ll T$.

We are not the first to consider a similar learning problem, although most prior works do require some form of external supervision, which we do not use. Here, however, we consider an additional key challenge that the images $\mathbf{I}_{0:T_0}(\mathcal{R})$ *do not* contain sufficient information to successfully predict the long-term objects' motion. This is because these few frames tell us nothing about the *nature* of the obstacles in the scenario. In fact, under the assumptions that obstacles of type $B$, $A$ and $U$ have similar or even the identical appearance, it is not possible to predict whether the ball will bounce, move above, or move under any such obstacle. This situation is representative of an agent that needs to operate in a new complex environment and must learn more about it, from past experiences, before it can do so reliably.

Thus, we consider a modification of the setup described above in which the model can experience each new scenario for a while, by observing the motion of the ball, before making its own predictions.

Formally, an *experience* is a collection of $N$ runs $\mathcal{E} = (\mathcal{R}_1, \ldots, \mathcal{R}_N)$ all relative to the same scenario $\mathcal{S}$ with each a different trajectories $\mathbf{y}_1, \ldots, \mathbf{y}_N$. By observing such examples, the model must determine the nature of the obstacles and then use this information to correctly predict the motion of the ball in the future. We cast this as the problem of learning a mapping

$$\Omega \,:\, (\mathbf{I}_{0:T_0}(\mathcal{R}), \mathbf{I}(\mathcal{E})) \,\longmapsto\, \mathbf{I}(\mathcal{R}), \tag{1}$$

where $\mathbf{I}(\mathcal{E}) = (\mathbf{I}(\mathcal{R}_1), \ldots, \mathbf{I}(\mathcal{R}_N))$ are the videos corresponding to the runs in the experience. We will call $\mathcal{R}$ the *prediction run* to distinguish it from the experience runs $\mathcal{E}$.

## 3.1 A META-LEARNING PERSPECTIVE

The setup we have described above can be naturally described as meta-learning. Namely, eq. (1) can be thought of as incorporating a "local" learning rule $\mathcal{M}$ that maps the experience $\mathcal{E}$ to a scenario-specific predictor $\hat{\Omega}$ on the fly: $\hat{\Omega}(\cdot) = \Omega(\cdot, \mathbf{I}(\mathcal{E})) = \mathcal{M}[\mathbf{I}(\mathcal{E})]$. Hence $\mathcal{M}$ must extract from the experience as much information as possible about the underlying scenario and transfer it to the scenario-specific predictor $\hat{\Omega}$. In order to learn $\mathcal{M}$, we consider *meta-samples* of the type $(\mathcal{S}, \mathcal{R}, \mathcal{E})$ comprising a scenario $\mathcal{S}$, a prediction run $\mathcal{R}$ and $N$ experience runs $\mathcal{E}$. Given a dataset $\mathcal{D}$ of such meta-samples, meta-learning searches for the mapping $\mathcal{M}$ that minimizes the error on the prediction runs:

$$\mathcal{M}^* = \operatorname*{argmin}_{\mathcal{M}} \frac{1}{|\mathcal{D}|} \sum_{(\mathcal{S},\mathcal{R},\mathcal{E}) \in \mathcal{D}} \ell(\mathbf{I}(\mathcal{R}), \hat{\Omega}(\mathbf{I}_{0:T_0}(\mathcal{R}))), \quad \hat{\Omega} = \mathcal{M}[\mathbf{I}(\mathcal{E})]. \tag{2}$$

## 3.2 COMPRESSED DYNAMIC EXPERIENCES

Concretely, we parameterise the scenario-specific predictor $\hat{\Omega}(\cdot; \boldsymbol{w}, \boldsymbol{m})$ using parameters $\boldsymbol{w}$, the weights a deep neural network, which is fixed and scenario-independent, and $\boldsymbol{m}$, which is scenario-specific. The latter is extracted by a network $\boldsymbol{m} = \Psi_1(\mathbf{I}(\mathcal{E}))$ from the experience videos $\mathbf{I}(\mathcal{E})$. Since we expect $\boldsymbol{m}$ to provide information about the nature of the obstacles in the scenario, we let $\boldsymbol{m} \in \mathbb{R}^{1 \times H \times W}$ be a tensor with the same spatial resolution as the scenario and interpret it as an "obstacle mask".

Given that the function $\Psi_1$ takes as input a number of video sequences, one may think to implement it as a recurrent neural network; however, recurrent networks scale badly, especially in our meta-learning context — to the point that we had difficulties in even running such a model at a non-trivial scale. Instead, we propose to construct $\Psi_1$ based on a compact representation of the experience which leads to a much more efficient design.

For this, we use the concept of *dynamic image* (Bilen et al., 2016), which encodes a video as a weighted average of its frames: $f_1(\mathbf{I}_0, \ldots, \mathbf{I}_{T-1}) = \sum_{t=0}^{T-1} \alpha_t \mathbf{I}_t$ with $\alpha_t = \sum_{i=t}^{T-1} \frac{2(i+1)-T-1}{i+1}$. Since the dynamic image is only sensitive to changes in the video, we complement it by computing also the *median image* $f_2(\mathbf{I}_0, \ldots, \mathbf{I}_{T-1}) = \operatorname{median}_{t=0,\ldots,T-1} \mathbf{I}_t$ and combine the two functions $f_1$ and $f_2$ in a single representation $f$ by stacking the respective outputs along the channel dimension. With this design, we can rewrite the map $\Psi_1$ as follows:

$$\boldsymbol{m} = \Psi_1(\mathbf{I}(\mathcal{E})) = \operatorname*{maxpool}_{n=1,\ldots,N} \hat{\Psi}_1(f(\mathbf{I}(\mathcal{R}_n))). \tag{3}$$

Here the map $\hat{\Psi}_1 : \mathbb{R}^{6 \times H \times W} \to \mathbb{R}^{1 \times H \times W}$, which can be implemented as a standard CNN, takes as input the dynamic/median image and produces as output the obstacle mask $\boldsymbol{m}$. The pooling operator summarizes the information extracted from multiple runs into a single mask $\boldsymbol{m}$.

We also consider a second similar map $\hat{\Psi}_2 : \mathbb{R}^{6 \times H \times W} \to \mathbb{R}^{C \times H \times W}$ to extract an "appearance" tensor $\boldsymbol{a}$. The latter helps the generator to render obstacles above or underneath the moving objects as needed. The response of this function is also max-pooled over runs, but channel-wise: $[\operatorname{pool}_{k=1,\ldots,N} \boldsymbol{a}_k]_{cvu} = [\boldsymbol{a}_{k(c)}]_{cvu}$ where $k(c) = \operatorname{argmax}_{k=1,\ldots,N} \sum_{vu} [\boldsymbol{a}_k]_{cvu}^2$.

In practice, $\hat{\Psi}_1$ and $\hat{\Psi}_2$ are implemented as a single neural network $\Psi : \mathbb{R}^{6 \times H \times W} \to \mathbb{R}^{(1+C) \times H \times W}$ where $\hat{\Psi}_1$ is the first output channel and $\hat{\Psi}_2$ the others.

**Optional obstacle mask supervision.** In the experiments, we show that the map $\hat{\Psi}$ can be learned automatically without any external supervision. We contrast this with supervising $\hat{\Psi}_1$ with an oracle

rendition of the obstacle map. To this end, we define the tensor $\boldsymbol{m}_{\text{gt}}$ for a scenario $\mathcal{S}$ to be the indicator mask of whether a pixel contains a solid obstacles (including the perimetral walls) and then minimize the auxiliary loss $\ell(\boldsymbol{m}_{\text{gt}}, \Psi_1(\mathbf{I}(\mathcal{E}))) = \|\boldsymbol{m}_{\text{gt}} - \Psi_1(\mathbf{I}(\mathcal{E}))\|^2$.

### 3.3 AUTO-REGRESSIVE PREDICTOR

The predictor $\Phi$ is designed as an RNN that takes as input $T_0$ past *states* $\mathbf{x}_{t-T_0:t} = (\mathbf{x}_{t-T_0}, \ldots, \mathbf{x}_{t-1})$ and outputs a new state $\mathbf{x}_t$. Each state variable is in turn a distributed representation of the physical state of the system, in the form of a "heatmap" $\mathbf{x}_t \in \mathbb{R}^{H \times W}$. Considering $T_0$ past states allows the model to auto-regressively represent the dynamics of the system if so learning chooses to do.

The predictor also takes as input the scenario representation $\boldsymbol{m}$ given by eq. (3). The first $T_0$ state variables are initialized from observations $\mathbf{I}_{0:T_0}(\mathcal{R})$ via an initialization function $\eta : \mathbb{R}^{3 \times H \times W} \to \mathbb{R}^{H \times W}$. We thus have

$$\mathbf{x}_{0:T_0} = [\eta(\mathbf{I}_t(\mathcal{R}))]_{t=0:T_0} \qquad \text{(initialization)} \qquad (4)$$
$$\mathbf{x}_t = \Phi(\mathbf{x}_{t-T_0:t}, \boldsymbol{m}) \qquad \text{(auto-regressive prediction)} \qquad (5)$$

In short, this model estimates recursively the evolution of the system dynamics from the visual observations of the first $T_0$ samples.

**Conditional generator.** Variable $\mathbf{x}_t$ contains information about the state of the moving objects (balls). This is then converted into a prediction of the corresponding video frame, combining also the appearance tensor $\boldsymbol{a}$ and the median of the first $T_0$ images in the sequence $\hat{\mathbf{I}} = \text{median}_{t=0,\ldots,T_0-1} \mathbf{I}_t$. This is formulated as a conditional generator network $\hat{\mathbf{I}}_t = g(\mathbf{x}_t, \boldsymbol{a}, \hat{\mathbf{I}})$.

**Video reconstruction loss.** Next, we discuss the nature of the loss equation 2. Owing also to the static background, the *conditional* video generation task is relatively simple provided that the dynamics are estimated correctly. As a consequence, the generated videos are likely to closely approximate the ground truth ones, so the loss function $\ell$ in equation 2 does not require to be complex. In our experiments, we combine the $L^2$ image loss with the *perceptual* loss of (Johnson et al., 2016). The latter is obtained by passing the ground-truth and predicted images through the first few layers of a pre-trained deep neural network $e$, VGG-16 (Simonyan & Zisserman, 2015) in our case, and then comparing the resulting encodings in $L^2$ distance. The overall loss is then given by $\ell(\mathbf{I}_t, \hat{\mathbf{I}}_t) = \lambda_\mathbf{I}\|\mathbf{I}_t - \hat{\mathbf{I}}_t\|^2 + \lambda_p\|e(\mathbf{I}_t) - e(\hat{\mathbf{I}}_t)\|^2$ (details in the sup. mat.). The perceptual loss is robust to small shifts in the image reconstructions that may arise from imperfect physics predictions and favors reconstructing a sharp image; by comparison, the $L^2$ loss alone would result in blurrier images by regressing to the mean. In practice, trading-off the $L^2$ and perceptual losses results in qualitatively better images as well as in better trajectory prediction.

**State space integrity.** An issue with the recursion equation 5 is that the state $\mathbf{x}_t$ may, in the long run, falls outside the space of meaningful states, leading to a catastrophic failure of the predictor. This is especially true for predictions longer than the ones used for training the model. In order to encourage the recursion to maintain the integrity of the state over time, we add a self-consistency loss $\|\mathbf{x}_t - \eta(\mathbf{I}_t(\mathcal{R}))\|^2$. Here the network $\eta$ is the same that is used to extract the state space from the initial frames of the video sequence in equation 4 which weights are fixed. In practice, we find that, by using this additional loss, our recursion learns to maintain the integrity of the state nearly indefinitely.

## 4 EXPERIMENTAL SETUP

**Data.** A scenario $\mathcal{S}$ (see embedded figure) is generated by sampling a certain number of obstacles of different types $m_i$ and shapes $O_i$ (either rectangular or custom), placing them at random locations and orientations on a 2D board. We consider boards with either two rectangular obstacles (denoted R2), a random number from 3 to 4 rectangular obstacles (R4) or two curved obstacles (C). Scenarios are rendered by painting the background and a wall around it. Then, all obstacle are painted in a solid colour, randomly picked from a fixed palette to ensure sufficient contrast. The background is also painted in solid color (general case) or by using a texture image (only curved shapes, denoted C+T). Crucially, there is no correlation between an obstacle's type and its shape, location and color, so its type cannot be inferred by appearance alone.

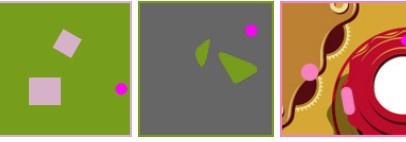

Runs $\mathcal{R}$ are generated by placing one or more dynamic objects ("balls") with a random initial location and a random momentum oriented towards the obstacles, simulating the 2.1D physics, and rendering the balls as circles with a fixed color (see fig. A1 in supp. mat. for more examples).

For each scenario $\mathcal{S}$, we sample $N + 1$ runs, using the first as a prediction run and the others as experience runs, forming triplets $(\mathcal{S}, \mathcal{R}, \mathcal{E})$. Unless otherwise stated, we set $N = 7$ and let each run evolve for 60 frames, including for the experience runs. Thanks to our compact representation, we only use 200MB for 7 experiences for backpropagation during training. Using a recurrent network requires an amount of memory larger in proportion with the length of an experience, i.e. 60-fold, so 12GB for 64×64 boards, which wouldn't allow the reccurrent architecture to fit into memory. Using shorter runs is also not possible, as they tend to contain no collisions.

For training, samples $(\mathcal{S}, \mathcal{R}, \mathcal{E})$ are drawn in an on-line fashion and are thus unlimited. For testing, we use a fixed set of 200 samples. We generate boards of size $64 \times 64$ and $128 \times 128$, but use the latter only for testing purposes to evaluate generalization. Unless otherwise stated, the board size is $64 \times 64$.

**Evaluation metrics.** We report the video prediction error as average $L_2$ image loss. In order to assess the ability of the method to predict good trajectories, we note that blobs tend to emerge in the heatmaps $\mathbf{x}_t$ in correspondence of the tracked objects. Hence, we handcrafted a simple blob detector $h(\mathbf{x}_t)$ that detects the blobs contained in the heatmaps $\mathbf{x}_t$ (see sup. mat.). We then report the number of detected blobs vs the actual number of moving objects and, for each, the distance in pixel space of the predicted blob center and ground-truth object center, averaged over the different scenarios. For each experiment we report mean and standard deviation across all sampled scenarios.

**Baselines.** We compare with three baselines. The first is a version of the Interaction Networks (Battaglia et al., 2016), trained with perfect knowledge of the object locations, background and object generation. The baseline works directly on the object positions and regresses object positions. The ground truth obstacle map is given as input and transformed into a vector thanks to a pre-trained VGG-16 architecture. We also reimplemented the RLVN network of Wang et al. (2018), which has a vector representation bottleneck, and compared video prediction error results. Note that this baseline wouldn't adapt to bigger board. Our last baseline amounts to running the ground-truth physics simulator after removing the obstacles from the board (and thus results in perfect predictions for trajectories that do not hit solid obstacles).

**Implementation details.** Networks $\Phi$ and $\eta$ share very similar auto-encoder type architecture, the network $\Psi$ uses a U-Net-like architecture (Ronneberger & Fischer, 2015). As stated in Wang et al. (2018) intuitively $\Psi$ would better preserve appearance while the structure of $\Phi$ would loose some appearance structure and put an inductive bias on dynamic predictions. $g$ is a fully-convolutional 6-layers stack. Our implementation uses TensorFlow 1.9 (Abadi et al., 2015) and the Adam optimizer (Kingma & Ba, 2014) with learning rate $10^{-4}$ and Xavier initialization (Glorot & Bengio, 2010). We used a batch size of 10 samples $(\mathcal{S}, \mathcal{R}, \mathcal{E})$. Models are first trained for 110,000 iterations using only the $L^2$ and self-consistency losses for eq. (2), and then optionally fine-tuning for further 1,000 iterations using the perceptual loss as well. All models are trained on R2 with board size $64^2$ and tested *without any fine-tuning* on different board types and sizes (see Tab. 1,2,3,A.3). For the C+T scenarios, we first train the model using flat colour R2 scenarios and then fine-tune for 55,000 iterations using the textured data. Unless otherwise specified, models are trained in a fully unsupervised fashion. Full details can be found in the sup. mat.

## 5 RESULTS

**Full system.** Table 1 rows 1-4 report the prediction accuracy of our system on the 200 simulated test sequences using one moving object. In the table, we distinguish three durations: $T_0 = 4$ is the number of frames to bootstrap the prediction of a test runs, $T_{\text{train}} = 20$ is the length of prediction runs observed during meta-training and $T_{\text{test}} = kT_{\text{train}}$ is the length of the runs observed during meta-testing, where $k = 1, 3, 5$. We test both $64 \times 64$ boards (which is the same size used for training) and larger $128 \times 128$ boards to test spatial generalization. We also compare training using the $L^2$ or the $L^2$+perceptual loss for video prediction supervision. We report the number of detected objects (which should be 1 in this experiment), the video prediction error, and the position prediction error.

Table 1: **Predicting one moving object.** Obst. is the obstacle type (R2, R4, C). The test board size can be either $64 \times 64$ (same as in training) or $128 \times 128$ and we consider using either no supervision or obstacle supervision. We test the average prediction error at $T_{\text{test}} = 20, 60, 100$ well above the duration $T_{\text{train}} = 20$ observed during training. Position errors were normalized by the board size diagonal. Video errors were normalized to board size $64^2$ where larger. $L^2$ loss on positions is used for (14)-(15).

| No. | Obst. | Sup. | Test brd. size | Train loss | $T_{\text{test}} = T_{\text{train}} = 20$ | | | $T_{\text{test}} = 3 \times T_{\text{train}}$ | | | $T_{\text{test}} = 5 \times T_{\text{train}}$ | | |
|---|---|---|---|---|---|---|---|---|---|---|---|---|---|
| | | | | | # obj. | Vid. $L_2$ | Pos. err. | # obj. | Vid. $L_2$ | Pos. err. | # obj. | Vid. $L_2$ | Pos. err. |
| (1) | R2 | None | $64^2$ | $L^2$ | 1.0±0.2 | 2.5±2.6 | .036±.096 | 0.3±0.5 | 4.6±3.3 | .404±.241 | 0.1±0.3 | 4.7±3.3 | .526±.180 |
| (2) | R2 | None | $128^2$ | $L^2$ | 0.7±0.4 | 1.7±1.3 | .179±.236 | 0.1±0.3 | 1.7±1.1 | .484±.228 | 0.0±0.1 | 1.7±1.1 | .531±.209 |
| (3) | R2 | None | $64^2$ | Percep. | 1.0±0.2 | 2.5±2.9 | .028±.073 | 1.0±0.2 | 5.3±4.0 | .145±.148 | 1.0±0.2 | 5.5±4.0 | .286±.168 |
| (4) | R2 | None | $128^2$ | Percep. | 0.9±0.4 | 1.7±1.4 | .104±.170 | 1.0±0.4 | 2.0±1.4 | .227±.179 | 1.0±0.4 | 2.0±1.4 | .320±.172 |
| (5) | R2 | Obst. | $64^2$ | Percep. | 1.0±0.1 | 3.0±3.3 | .018±.023 | 1.0±0.2 | 5.7±4.4 | .112±.086 | 1.0±0.3 | 5.8±4.2 | .231±.141 |
| (6) | R2 | Obst. | $128^2$ | Percep. | 8.3±3.3 | 2.4±1.9 | .323±.064 | 26.5±6.4 | 5.6±4.0 | .355±.054 | 25.8±6.3 | 5.8±4.1 | .363±.133 |
| (7) | C | None | $64^2$ | Percep. | 1.0±0.1 | 2.9±3.5 | .040±.086 | 1.1±0.3 | 5.4±4.1 | .175±.160 | 1.0±0.3 | 5.4±4.0 | .265±.155 |
| (8) | C | None | $128^2$ | Percep. | 0.9±0.4 | 1.7±1.4 | .104±.170 | 1.0±0.4 | 2.0±1.4 | .227±.179 | 1.0±0.4 | 5.6±4.6 | .320±.172 |
| (9) | C | Obst | $64^2$ | Percep. | 1.0±0.1 | 2.9±3.1 | .024±.072 | 1.0±0.1 | 5.7±4.2 | .128±.114 | 1.0±0.2 | 5.5±4.1 | .247±.141 |
| (10) | C+T | None | $64^2$ | Percep. | 1.0±0.1 | 3.4±2.9 | .017±.044 | 1.0±0.2 | 6.1±3.4 | .094±.076 | 1.0±0.2 | 6.2±2.9 | .222±.147 |
| (11) | C+T | None | $128^2$ | Percep. | 1.0±0.1 | .675±.685 | .012±.033 | 1.0±0.2 | 1.4±.814 | .057±.065 | 0.9±0.3 | 1.4±.750 | .121±.088 |
| | | | | | **Ablation**: removing the median and dynamic images, respectively (see text) | | | | | | | | |
| (12) | R2 | None | $64^2$ | Percep. | 1.0±0.3 | 4.0±4.0 | .074±.115 | 0.8±0.5 | 5.1±4.0 | .302±.202 | 0.6±0.5 | 5.0±3.8 | .357±.212 |
| (13) | R2 | None | $64^2$ | Percep. | 1.4±0.6 | 3.2±3.4 | .054±.062 | 1.8±0.8 | 5.5±4.2 | .268±.151 | 1.7±0.8 | 5.6±4.2 | .326±.136 |
| | | | | | **Comparison**: Interaction Network (14-15), RLVN (16) and Ground Truth Simulator minus Obstacles (17-18) | | | | | | | | |
| (14) | R2 | Full | $128^2$ | $L^2$ Pos. | - | - | .038±.027 | - | - | .166±.088 | - | - | .331±.152 |
| (15) | C | Full | $128^2$ | $L^2$ Pos. | - | - | .038±.031 | - | - | .171±.095 | - | - | .349±.203 |
| (16) | R2 | - | $64^2$ | - | - | 3.6±3.2 | - | - | 5.3±3.3 | - | - | 5.5±3.5 | - |
| (17) | R2 | - | $64^2$ | - | - | - | .060±.099 | - | - | .229±.224 | - | - | .224±.209 |
| (18) | C | - | $64^2$ | - | - | - | .055±.099 | - | - | .219±.217 | - | - | .232±.228 |

We find that switching from the $L^2$ to the $L^2$+perceptual loss for training (rows 1 vs 3 and 2 vs 4) reduces significantly the trajectory prediction error and prevents the object from disappearing. This, however, results to slightly larger $L_2$ video prediction error compared to empty boards predicted in the long run by models trained with $L^2$ loss only. We also find that generalization through time is excellent: prediction errors accumulate over time (as it is unavoidable), but the recursion evolves without collapsing well beyond $T_{\text{train}}$ (qualitatively we found that the recursion can be applied indefinitely). This should be contrasted with prior work on intuitive physics (Fragkiadaki et al., 2016) where recursion was found to be unstable in the long term.

Spatial generalization to larger boards (rows 2,4,6) is also satisfactory if the perceptual loss is used. However, we did notice the emergence of a few artifacts in this case.

In rows 5-6 we use external supervision to train the obstacle map predictor as suggested in section 3.2 (the rest of the system is still unsupervised). Supervision improves the trajectory prediction accuracy for boards of the same size as the ones seen in training. However, generalization to larger boards is poorer, suggesting that explicit supervision causes the model to overfit somewhat.

Rows 7-11 show results for more complex shapes (C) and appearances (C+T). Prediction accuracy is only slightly worse than with the simpler obstacles (e.g. rows 3 vs 7 or 9 vs 10).

Table 2: **Importance of experience.** For the unsupervised model, we report the trajectory prediction error at $T = 40$ (rows 1-3) and the obstacle mask $L_2$ prediction error (rows 4-6) for different obstacle types. The number of runs $N$ in the experiences is varied from 0 to 50. For the obstacle mask prediction error, we also report the trivial baseline $\|\boldsymbol{m}_{\text{gt}} - \boldsymbol{m}_{\text{all-on}}\|_2$ as 'All-ON' (see text). Position errors were normalized by the board size ($64^2$) diagonal. $T_{\text{train}} = 20$.

| No | Err. | Obst. | All-ON | $N = 0$ | 1 | 2 | 5 | 7 | 10 | 20 | 50 |
|---|---|---|---|---|---|---|---|---|---|---|---|
| (1) | Pos. | R2 | — | .462±.136 | .144±.157 | .108±.136 | .083±.128 | .077±.120 | .071±.114 | .070±.116 | .070±.104 |
| (2) | Pos. | R4 | — | .502±.151 | .189±.169 | .158±.164 | .122±.155 | .110±.149 | .114±.130 | .110±.154 | .114±.152 |
| (3) | Pos. | C | — | .459±.124 | .161±.172 | .125±.158 | .099±.139 | .099±.146 | .099±.144 | .101±.144 | .106±.143 |
| (4) | Obs. | R2 | 13.2±11.1 | 24.6±16.4 | 10.3±9.5 | 7.6±8.4 | 4.2±5.4 | 3.8±4.7 | 3.5±3.3 | 3.6±3.5 | 3.8±3.7 |
| (6) | Obs. | R4 | 16.8±13.3 | 23.6±15.1 | 14.1±10.9 | 11.5±9.9 | 7.7±8.0 | 7.4±7.3 | 7.3±7.2 | 7.4±7.2 | 8.4±8.3 |
| (5) | Obs. | C | 9.4±8.9 | 27.8±15.8 | 10.1±8.2 | 7.0±7.0 | 5.4±5.6 | 5.3±5.2 | 5.1±5.2 | 5.7±5.6 | 6.0±5.9 |

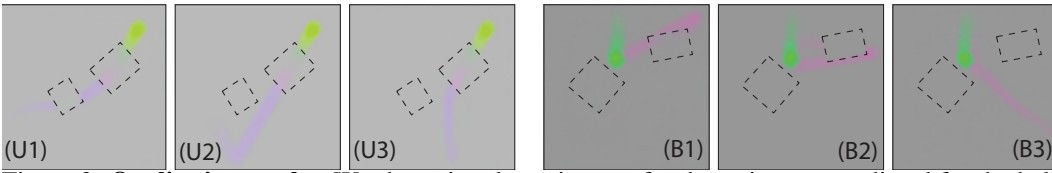

Figure 3: **Qualitative results**. We show time lapse images for the trajectory predicted for the ball colliding with obstacles of type U (permeable; left) and B (solid; right). We test three variants of the model: (1) the model supervised with ground-truth obstacle masks, (2) the unsupervised model and (3) the model after suppressing the dynamic image that summarizes the experience (ablation). Qualitatively, (1,2) produce plausible predictions but (3) does not.

**Ablation.** We suppress respectively the median image (row 12) and dynamic image (row 13) in the feature extractor $f$ summarizing the experiences. We notice a sensible deterioration of performance compared to row 3, suggesting that both components are important.

**Comparison.** Compared to the fully-supervised IN (rows 14-15), our unsupervised method is highly competitive, especially for longer simulation times (rows 4,8). The method also outperforms (except for very long simulations where drift prevails) the ground-truth simulator that ignores the obstacles (rows 16,17). Finally we show that our method surpasses RLVN (Wang et al., 2018)(row 16), qualitatively, this method produces similar results to (13) and the object tends to disappear when encounting an object.

**The importance of experience.** We investigate how much information the model extracts from the experiences $\mathcal{E}$ by ablating the number of runs $N$ in it. For the special case $N = 0$, since there are no runs, we generate an pseudo experience run by copying the first frame $\mathbf{I}_0(\mathcal{R})$ of the prediction run $\mathcal{R}$.

In table 2, we report the trajectory prediction accuracy for different obstacle types (row 1-3). We also test the ability to predict the obstacle mask $\boldsymbol{m}_{\text{gt}}$ (defined in section 3). Since the model is unsupervised, the learned mask $\boldsymbol{m}$ will not in general be an exact match of $\boldsymbol{m}_{\text{gt}}$ (e.g. the mask range is arbitrary); hence, after fixing all the parameters in the model, we train a simple regression network that maps $\boldsymbol{m}$ to $\boldsymbol{m}_{\text{gt}}$ and report the prediction accuracy of the latter on the test set (see sup. mat. for details). As a point of comparison, we also report the error $\|\boldsymbol{m}_{\text{gt}} - \boldsymbol{m}_{\text{all-on}}\|_2$ of the trivial baseline that predict a mask $\boldsymbol{m}_{\text{all-on}}$ where all objects are highlighted, regardless of their flag.

We note that there is a very large improvement when we move from zero to one experience runs, and a smaller but non-negligible improvement until $N = 10$. Furthermore, comparing the results with predicting $\boldsymbol{m}_{\text{all-on}}$ shows that the system can tell from the experiences which obstacles are solid and which are not.

Table 3: **Predicting the obstacles: supervision vs no supervision.** The table reports the obstacle mask prediction error ($L_2$) for a network trained with or without supervision for the obstacles (see text). The last column shows a multi-ball predictor operated with 3 balls in each run. $T_{\text{train}} = 20$.

| Supervision | R2 | R4 | C | C+T | R2 (3 balls) | C+T $\rightarrow$ R2 |
|---|---|---|---|---|---|---|
| Obstacle | $2.7_{\pm 4.4}$ | $6.3_{\pm 7.8}$ | $3.5_{\pm 4.9}$ | $2.6_{\pm 3.9}$ | $2.6_{\pm 3.9}$ | $7.1_{\pm 9.2}$ |
| None | $3.8_{\pm 4.7}$ | $7.4_{\pm 7.3}$ | $5.3_{\pm 5.2}$ | $4.0_{\pm 5.2}$ | $3.9_{\pm 3.5}$ | $7.6_{\pm 8.9}$ |

**Supervised obstacle regression.** Table 3 compares our unsupervised method to using full obstacle map supervision in order to predict the obstacle map $\boldsymbol{m}_{\text{gt}}$ from the experiences $\mathcal{E}$. For the unsupervised system, the obstacle map is estimated as explained in the paragraph above. As expected, supervised learning achieves a lower error, but the unsupervised method is still much better than the trivial baseline of table 2. Finally, we observe that for network fine-tuned on C+T and evaluated back on R2 (C+T$\rightarrow$R2 in Table 3) obstacle predictions results remain competitive with results on R2 only while object appearance and backgrounds drastically changed in the new setting.

**Multiple moving objects.** We also test whether the system trained with a single moving object can generalize to multiple ones. In table A.3 (see sup. mat.) we show that the network can simulate the motion properly until balls collide, after which they merge (see videos in the sup. mat.). This indicates that, just as physical laws, the rules learned by the model are local. We also train a model (scenario R2) showing it from 1 to 3 balls in each run. This model is able not only to correctly handle ball collisions, but is also able to generalize correctly to several more objects on the board. Furthermore,

as shown in table 3, this model still predicts correctly the obstacle masks from experiences, despite the act that the latter are much more crowded.

## 6 CONCLUSIONS

We have demonstrated a system that can learn an intuitive model of physics in an unsupervised manner. Differently from most prior works, our system is also able to learn on-the-fly and very efficiently some physical parameters of new scenarios from a few experiences. Prediction results are strong, competitive with fully-supervised models, and predictors generalize well over time space, and an arbitrary number of moving objects. Our next challenge is to apply the system to more difficult scenarios, including non-frontal views of the physical worlds. For this, we plan to learn a function to summarize past experiences in a more general manner than the dynamic image could.

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

## A1    DATA GENERATION

In fig. A1 and fig. A2 we provide more details on the scene aspect. For solid background color, the objects and background colours are sampled to be different. Every rectangular object have height and width randomly sample in between 10 and 17 pixels. Custom object are loaded from template images and randomly scaled from 1 to 2.

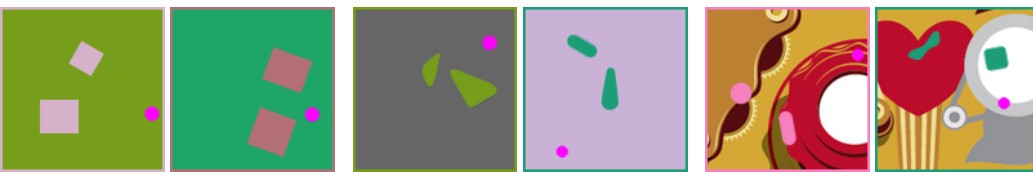

Figure A1: **Dataset samples.** Pairs of sample data from left to right: R2, C, C+T.

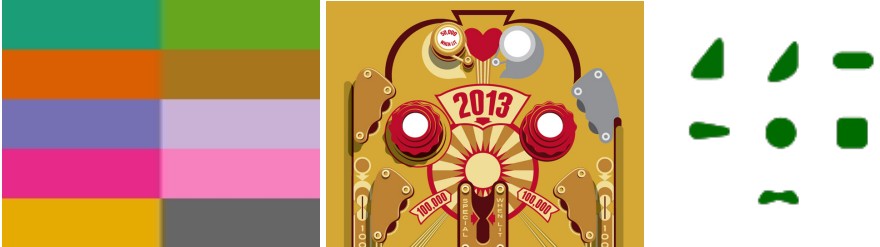

Figure A2: **Additional material used for data generation**. From left to right: color palette for solid object/background color (R2,R4,C), texture used to increase background complexity in (C+T), set of custom objects for (C) data generation.

## A2    IMPLEMENTATION DETAILS

### A2.1    NETWORK ARCHITECTURE

In fig. A3 we show the details of all the networks of our pipeline. The regression network mentionned in sec.5 is a simple 3 layers fully convolutional network with [3x3] kernel convolution and intermediate channel of size 64.

### A2.2    TRAINING DETAILS

In the overall loss $l$ is:
$$\ell(t) = \lambda_{\mathbf{I}}\|\mathbf{I}_t - \hat{\mathbf{I}}_t\|^2 + \lambda_{\mathbf{x}}\|\mathbf{x}_t - \eta(\mathbf{I}_t(\mathcal{R}))\|^2 + \lambda_p\|e(\mathbf{I}_t) - e(\hat{\mathbf{I}}_t)\|^2$$
With $t$ ranging from 0 to $T_{train}$. For $t = \{0, .., T_0\}$ we used $\lambda_{\mathbf{I}} = 1$ and $\lambda_{\mathbf{x}} = \lambda_{\mathbf{p}} = 0$. In such case $\hat{I}_t = g(x_t, \mathbf{a}, \hat{\mathbf{I}}) = g(\eta(\mathbf{I}_t), \mathbf{a}, \hat{\mathbf{I}})$.

For $t > T_0$, when training with $L^2$ only we used $\lambda_{\mathbf{I}} = \lambda_{\mathbf{x}} = 1$ and $\lambda_p = 0$. When finetuning with perceptual we weighted the perceptual loss with coefficient $\lambda_p = 10$ and the other losses with $\lambda_{\mathbf{I}} = \lambda_{\mathbf{x}} = 0.01$. Finally, perceptual loss uses feature extracted from conv3 of VGG-16.

The Interaction Network baseline is a simple Interaction Network to which state we concatenated the background feature extracted from a pre-trained VGG-16 network. The state propagator uses the last 4 states. The network is then trained over $T_{train} = 40$ time steps with Adam optimizer with learning rate $10^{-4}$ and batch size 50. We found that 50 000 iterations were sufficient to reach convergence.

All experiments are run using single NVIDIA-GPU Titan X.

### A2.3    EVALUATION DETAILS

To detect blob create a binary image by manually thresholding the intermediate heatmap $\mathbf{x}_t$ and using a simple Hough circle detector. Every threshold coefficient is hand picked for every model but kept constant across scenarios and different data.

For position errors we used an $L_2$ loss, when more than one object was detected we took the distance to the closest detected object and applied a fix penalty to the loss for all other detection (half the board size).

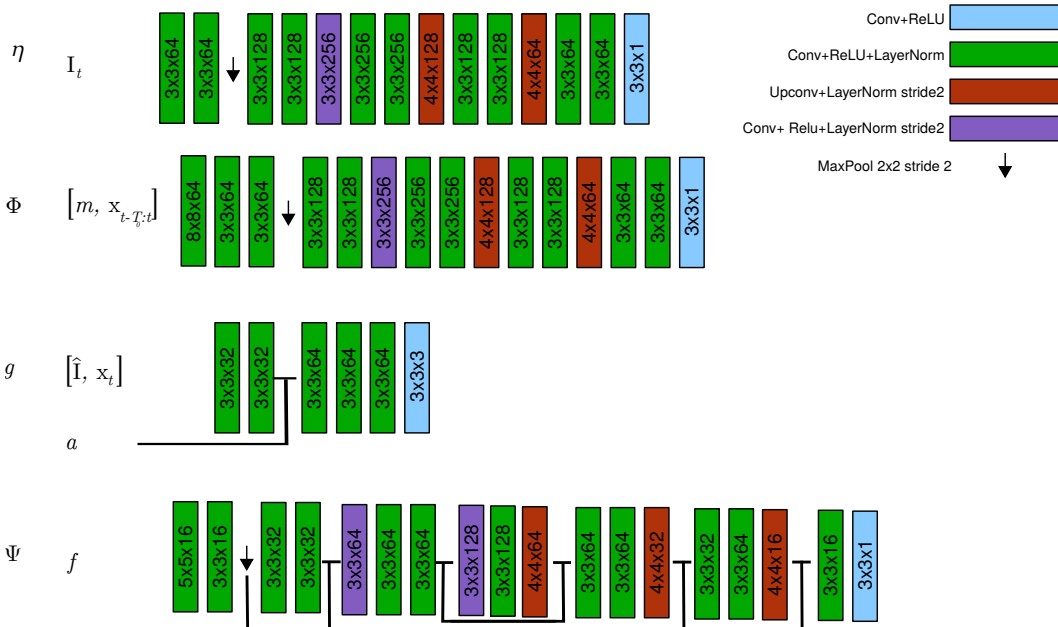

Figure A3: **Networks architecture detail**. All the networks generally uses layer normalization $\eta$ and $\Phi$ are simple auto-encoder type architecture while $\Psi$ shares similarity with U-Net.

Table A1: **Predicting one moving object.** Obstacle type is R2. The test board size is $64 \times 64$ We test the average prediction error at $T_{\text{test}} = 20, 60, 100$ well above the duration $T_{\text{train}} = 20$ observed during training. Position errors were normalized by the board size diagonal.

| No. | Sup. | Train loss | $T_{\text{test}} = T_{\text{train}} = 20$ | | | $T_{\text{test}} = 3 \times T_{\text{train}}$ | | | $T_{\text{test}} = 5 \times T_{\text{train}}$ | | |
|---|---|---|---|---|---|---|---|---|---|---|---|
| | | | # obj. | Vid. $L_2$ | Pos. err. | # obj. | Vid. $L_2$ | Pos. err. | # obj. | Vid. $L_2$ | Pos. err. |
| (1) | Oracle | Perc. | 1.0±0.1 | 2.5±2.9 | .013±.012 | 1.0±0.1 | 5.8±4.4 | .095±.095 | 1.0±0.3 | 6.2±4.5 | .201±.136 |
| (2) | None | Perc. | 1.0±0.1 | 3.2±3.1 | .030±.068 | 1.0±0.2 | 5.5±4.0 | .136±.151 | 0.9±0.4 | 5.5±4.0 | .262±.180 |

## A3 ADDITIONAL RESULTS

In Table A1 we report two more results on one moving object prediction. (1) is an upper bound of our system had full knowledge of $\{B, U\}$ obstacle locations at training time. We simply replaced $\boldsymbol{m}$ with the binary map of $B$ obstacles and $\boldsymbol{a}$ with the binary map of $U$ obstacles where 1s indicate the presence of the obstacle type. In this scenario we note although that we obtain an upper bound of the results introduced in Table 1, our method still remains competitive with this upper bound. (2) reports results of the network trained on C+T and evaluated back on R2. As in Table 3 we see that the network still manage to make accurate predictions. Table A2 is an extension of Table 2 in the paper for the **supervised model** and Table A3 introduces results on multiple balls prediction.

Table A2: **Importance of experience.** For the **supervised** model, we report the trajectory prediction error at $T = 40$ (rows 1-3) and the obstacle map prediction error (rows 4-6) for different obstacle types. The number of runs $N$ in the experiences is varied from 0 to 50. Position errors were normalized by the board size diagonal. $T_{\text{train}} = 20$.

| No | Err. | Obst. | $N = 0$ | 1 | 2 | 5 | 7 | 10 | 20 | 50 |
|---|---|---|---|---|---|---|---|---|---|---|
| (1) | Pos. | R2 | 1.81±1.41 | .94±.975 | .53±.81 | .15±0.41 | .08±.21 | .07±.16 | .10±.24 | .19±.38 |
| (2) | Pos. | R4 | 3.28±1.94 | 2.22±1.67 | 1.56±1.40 | .63±.84 | .43±.68 | .37±.60 | .44±.61 | .83±.97 |
| (3) | Pos. | C | 1.05±0.91 | .58±.61 | .36±.49 | .15±.29 | .13±.26 | .12±.22 | .16±.26 | .24±.36 |
| (4) | Obs. | R2 | 4.1±4.0 | 3.6±3.8 | 3.1±3.4 | 2.3±2.5 | 2.3±2.4 | 2.3±2.4 | 2.3±2.7 | 2.6±3.0 |
| (5) | Obs. | R4 | 6.1±2.2 | 4.1±3.8 | 3.1±3.4 | 2.8±3.3 | 2.7±3.1 | 2.6±3.0 | 2.7±2.9 | 3.3±3.6 |
| (6) | Obs. | C | 3.9±3.9 | 3.3±3.7 | 2.8±3.1 | 2.4±2.7 | 2.4±2.8 | 2.3±2.6 | 2.4±2.6 | 2.5±2.9 |

Table A3: **Predicting multiple moving object.** Obstacle type is R2, all network were trained on the fully unsupervised with perceptual loss. Num. is the maxmium number of balls per run during training. For instance Num=3 means that every runs sampled contained one to three balls. The test board is size $64 \times 64$ We test the average prediction error at $T_{\text{test}} = 20, 60, 100$ well above the duration $T_{\text{train}} = 20$ observed during training.

| Num. | $T_{\text{test}} = T_{\text{train}} = 20$ | | $T_{\text{test}} = 3 \times T_{\text{train}}$ | | $T_{\text{test}} = 5 \times T_{\text{train}}$ | |
|---|---|---|---|---|---|---|
| | # obj. | Vid. $L_2$ | # obj. | Vid. $L_2$ | # obj. | Vid. $L_2$ |
| | | | 3 Balls | | | |
| 1 | 2.6±0.8 | 6.5±5.2 | 1.7±0.9 | 8.5±5.9 | 1.6±0.8 | 8.5±6.0 |
| 3 | 3.1±0.6 | 4.6±4.8 | 2.9±0.8 | 10.4±7.6 | 2.9±0.9 | 11.0±8.0 |
| | | | 5 Balls | | | |
| 1 | 3.3±1.0 | 8.9±6.6 | 2.0±1.0 | 10.2±7.4 | 3.1±0.6 | 1.8±.775 |
| 3 | 5.5±0.7 | 6.6±6.2 | 5.1±1.1 | 13.0±9.5 | 5.1±1.0 | 13.5±10.0 |

