# OpenReview forum: "Unsupervised Intuitive Physics from Past Experiences"
_ICLR.cc/2020/Conference — Reject_

### Official Review · AnonReviewer1 · 2019-10-13
**Official Blind Review #1**

**Rating:** 3

**Review:**


Paper summary:

The paper proposes a method to predict the future trajectory of a ball (or a set of balls) given the first few frames of the trajectory and a set of experience runs in the same environment. The model first learns to convert the set of images to a set of corresponding heatmaps that encode the location of the ball. Then, a recurrent network generates the future states given the first few locations and the output of a network that learns abstract information from the experience runs. The network is trained using reconstruction and perceptual similarity loss.

Paper strengths:

It is interesting that the model is able to learn from a set of experience runs. Learning abstractions from a set of experience runs is an interesting direction.

Paper weaknesses:

The paper has two main problems. (1) The model and the experiments are designed for a very simplistic scenario. I do not think that machinery is really needed to perform prediction in this simple setting. (2) The paper needs major re-rewriting. Some of the main parts of the paper are not clear. Please refer to the comments below for more details.

- It is unclear how the initialization function (Equation 4) is learned. It is very unlikely that the auto-encoder produces a heatmap of the object locations only based on the reconstruction loss that it receives at the end. This should be clarified.

- Equation 2 is very confusing. Is \phi-hat a function of experience I(E) or I_{0:T}(R). It seems \phi-hat produces an image that is compared with I(R), but it doesn't seem that M produces an image.

- The proposed machinery is overkill for the very simple experimental setup. A nearest neighbor baseline would probably perform as well. We can extract a set of frames in the experience runs that are closest to the first few frames using some simple distance functions. Then, we can predict the future movements by copying the rest of the trajectory from the experience run.

- The only part that seems unsupervised is the location of the ball, which can be easily obtained by image subtraction in this simple setting. What else is unsupervised in the proposed approach?

- The beginning of section 3.2 mentions w, but there is no mention of it afterwards.

- Why does the backprop for sixty 64x64 images require 12Gb of memory? It would be good to explain that.

- Regarding the evaluation, how does it know which prediction corresponds to the which groundtruth to compute the distance that is mentioned? The prediction can correspond to any of the balls since there is no explicit notion of object in the model.

Due to the issues mentioned above, especially extremely simple experimental setup and lack of clarity, I chose "Weak Reject".



**Experience Assessment:**

I have published one or two papers in this area.

**Review Assessment: Checking Correctness Of Derivations And Theory:**

I carefully checked the derivations and theory.

**Review Assessment: Checking Correctness Of Experiments:**

I carefully checked the experiments.

**Review Assessment: Thoroughness In Paper Reading:**

I read the paper thoroughly.

---

> ### Author Response · Authors · 2019-11-12
> **Answer to Reviewer 1 (1/2)**
>
> We thank the reviewer for helpful comments and suggestions.
>
> > (1) The model and the experiments are designed for a very simplistic scenario. I do not think that machinery is really needed to perform prediction in this simple setting.
>
> (This comment was already answered in response to R2 we copy the text here for the reviewer’s convenience)
>
> While our task may appear simple, it is, in fact, quite challenging due to the unsupervised and image-based nature of the task.  Our work should be put in context to related work in intuitive physics. Many papers in this domain work with no images at all (Chang et al 2017., Battaglia et al. 2017, Sanchez-Gonzalez et al 2018), with either similar or less complex scenarios. The ones that do use images (Fragkiadaki et al. 2016), assume the renderer to be known thus preventing the objects to disappear over time. Our method not only works with raw images -- just learning to predict the  motion of a single ball from raw pixels is non trivial -- but also considers obstacles and multiple balls, each of which is a challenge in its own right (especially when done from raw pixels with no supervision). Finally, we propose a whole new formulation in which the intuitive physics learner can probe new environments on the fly to acquire information about them (in our case the type of obstacles).
>
>
> > - It is unclear how the initialization function (Equation 4) is learned. It is very unlikely that the auto-encoder produces a heatmap of the object locations only based on the reconstruction loss that it receives at the end. This should be clarified.
>
> It is in fact possible to learn such map because the initialization function receives signal both from \Phi and from reconstruction loss at T={0, …, T_0} (This reconstruction loss is already in the paper (Fig 1 and page 6)). The loss at {0,..,T_0} is computed with the only output of \eta at t= {0,...,T_0} (eq (4) page 5). We further clarified this point in the updated version of the paper (Appendix A 2.2)
>
> > - Equation 2 is very confusing. Is \phi-hat a function of experience I(E) or I_{0:T}(R). It seems \phi-hat produces an image that is compared with I(R), but it doesn't seem that M produces an image.
>
> M is a meta learner which from past experiences produces a function phi-hat (autoregressor \phi +rendering function g see Fig1. top right). We have modified the paper to take into account this suggestion as \Phi (now \Omega) in eq 3 and Section 3.1 was different from the auto-regressor of Fig1 and eq 5.
>
> > - The proposed machinery is overkill for the very simple experimental setup. A nearest neighbor baseline would probably perform as well. We can extract a set of frames in the experience runs that are closest to the first few frames using some simple distance functions. Then, we can predict the future movements by copying the rest of the trajectory from the experience run.
>
> It is unclear how such baseline would perform in practice. Since most of the experiences are drawn independently with random starting position and velocity, such baseline would perform poorly and be unable to fulfil one goal of this work i.e to learn propagation and object-environment interactions. Furthermore, the proposed baseline would not be able to propagate the ball state further than 60 steps e.g the size of the observed experiments.
>
> > - The only part that seems unsupervised is the location of the ball, which can be easily obtained by image subtraction in this simple setting. What else is unsupervised in the proposed approach?
>
> Apart from the location of the ball, the classification of obstacles {A,U,B} and the auto-regressive model which learnt not only to propagate the ball but also to model it’s interaction with its environment were unsupervised. Since the location of the objects were also unsupervised it is also a challenging task to maintain a representation of the object over time.
>
> >- The beginning of section 3.2 mentions w, but there is no mention of it afterwards.
>
> ‘w’ refers to the weights of the auto-regressive predictor \Omega-hat which comprises the auto-regressive network \Phi and the renderer g. This was clarified in the updated version in Section 3.2.
>
> >- Why does the backprop for sixty 64x64 images require 12Gb of memory? It would be good to explain that.
>
> Our light-weighted network \Psi was using only 200Mb of memory for backpropagation (using a batch size of 10 and 7 experiments). Since all experiments were using a compact representation each of them could be summed up in only one image. Using an RNN instead of this compact representation would have resulted in an increase of memory footprint equivalent to the number of steps in each experiments e.g., 60.

---

> > ### Author Response · Authors · 2019-11-12
> > **Answer to Reviewer 1 (2/2)**
> >
> > >- Regarding the evaluation, how does it know which prediction corresponds to the which groundtruth to compute the distance that is mentioned? The prediction can correspond to any of the balls since there is no explicit notion of object in the model.
> >
> > When more objects were detected we took the distance to the closest object and applied an arbitrary fixed penalty for each of the other detected objects (half of the board size). This was clarified in the updated version of the paper (Appendix A 2.3).

---

### Official Review · AnonReviewer2 · 2019-10-16
**Official Blind Review #2**

**Rating:** 3

**Review:**


Summary
The paper proposes an architecture for few-shot video prediction in which a number of videos are summarized through global pooling operations and passed into a video predictor that learns to leverage them for adaptation, similar in spirit to the RNN-based meta-learning approaches such as Santoro’16, Duan’16. Due to global image and feature pooling operations, the proposed approach is computationally efficient. Proof-of-concept experiments are presented in a simple simulated physical prediction setting. It is claimed that the proposed model achieves generalization to longer sequences and larger board sizes, as well as a larger number of objects.

Decision
The work considers an interesting task and shows promise, but the paper lacks a substantial contribution at this time. I recommend weak reject. To improve the paper, I recommend considering harder prediction tasks and lifting some of the restrictions of the method.

Pros
The paper proposes and evaluates a model for few-shot video prediction, which is an interesting and well-motivated task.

Cons
1. The method is only evaluated on a toy task. While a model is proposed that can successfully solve the presented toy task, certain assumptions, such as averaging directly in the image space, may prevent the findings of this paper to scale to more complex tasks, which demands further investigation.

2. It is claimed that the method generalizes to longer sequences and larger board sizes, however, the performance of the method seems to quickly deteriorate in both settings. Authors present anecdotal evidence that the method generalizes to larger number of objects, however, quantitatively the results again are much inferior to the method trained on a larger number of objects. While the presented qualitative generalization findings are interesting, they seem limited in scope, especially when taking into account the simplicity of the considered data.

Minor comments
1. On page 3, the paper states “We are not the first to consider a similar learning problem, although most prior works do require some form of external supervision.” While there is a substantial body of literature that uses extra supervision, the following foundational works do not: Finn et al, 2016a,b; Fragkiadaki, 2016. In addition, “a similar learning problem” is considered without supervision in many papers on video prediction, including: Ranzato et al., 2014; Srivastava et al., 2015; Mathieu et al., 2015; Denton et al., 2017, 2018; Villegas et al., 2017, 2019; Wichers et al., 2018; Castrejon et al., 2019. While it might not be necessary to cite all of these as they do not necessarily apply their models to intuitive physics, the statement in the paper seems rather unsubstantiated.
2. A similar meta-learning prediction problem is addressed in Nagabandi’19a,b, which are not cited.
3. The paper is somewhat cluttered with notation, which overall hampers the flow for the reader. The paper would benefit from slight reorganization to improve the flow. This could be done e.g. be structuring the paper to first present the computational task that it addresses, namely few-shot video prediction, and how each component of the model works to address that problem, and only later go into the detail of how the data were collected and how each individual module was implemented


Santoro et al, Meta-learning with memory-augmented neural networks.
Duan et al, RL2: Fast reinforcement learning via slow reinforcement learning.
Nagabandi et al, LEARNING TO ADAPT IN DYNAMIC, REAL-WORLD ENVIRONMENTS THROUGH META-REINFORCEMENT LEARNING,
Nagabandi et al, DEEP ONLINE LEARNING VIA META-LEARNING: CONTINUAL ADAPTATION FOR MODEL-BASED RL


--------------------- Update 11.19 -----------------------

I appreciate the newly provided qualitative generalization results (in response to R3). However, it is clear that the network has trouble maintaining the exact number of spheres in the scene, especially after collisions or interactions with obstacles. While this generalization finding is undoubtedly interesting, I am not convinced it is enough for a publication in ICLR.

Furthermore, to support the authors' claim that the prediction performance on long sequences is satisfactory, I suggest comparison with a hand-crafted baseline, such as one suggested by R1, or one that makes use of explicit physics.


**Experience Assessment:**

I have published one or two papers in this area.

**Review Assessment: Checking Correctness Of Derivations And Theory:**

N/A

**Review Assessment: Checking Correctness Of Experiments:**

I carefully checked the experiments.

**Review Assessment: Thoroughness In Paper Reading:**

I read the paper thoroughly.

---

> ### Author Response · Authors · 2019-11-12
> **Answer to Reviewer 2 (1/2)**
>
> We thank the reviewer for helpful comments and suggestions.
>
> > 1. The method is only evaluated on a toy task.
>
> While our task may appear simple, it is, in fact, quite challenging due to the unsupervised and image-based nature of the task.  Our work should be put in context to related work in intuitive physics. Many papers in this domain work with no images at all (Chang et al 2017., Battaglia et al. 2017, Sanchez-Gonzalez et al 2018), with either similar or less complex scenarios.  The ones that do use images (Fragkiadaki et al. 2016), assume the renderer to be known thus preventing the objects to disappear over time. Our method not only works with raw images -- just learning to predict the  motion of a single ball from raw pixels is non trivial -- but also considers obstacles and multiple balls, each of which is a challenge in its own right (especially when done from raw pixels with no supervision). Finally, we propose a whole new formulation in which the intuitive physics learner can probe new environments on the fly to acquire information about them (in our case the type of obstacles).
>
>
>
> > While a model is proposed that can successfully solve the presented toy task, certain assumptions, such as averaging directly in the image space, may prevent the findings of this paper to scale to more complex tasks, which demands further investigation.
>
> We believe this approximation has already been verified on more complex tasks. In fact, averaging in image space was the simplification introduced by Bilen et al. 2016. and  showed good results on video classification tasks from real world data.
>
> > 2. It is claimed that the method generalizes to longer sequences and larger board sizes, however, the performance of the method seems to quickly deteriorate in both settings. Authors present anecdotal evidence that the method generalizes to larger number of objects, however, quantitatively the results again are much inferior to the method trained on a larger number of objects. While the presented qualitative generalization findings are interesting, they seem limited in scope, especially when taking into account the simplicity of the considered data.
>
> Since we integrate motion over time, errors accumulate linearly with time (also observed in Fragkiadaki et al. 2016, Chang et al. 2017). Furthermore, as the simulation duration increases, the ball bounces many more times, and each bounce increases uncertainty significantly. Hence, the goal here is to test whether the prediction deteriorates above what has to be expected due to the intrinsic ambiguity of the prediction task -- and it does not.
> Also note that we show that we can maintain the object representation over long sequences and larger board sizes, since the number of detected objects in both cases were found to be stable at around 1.

---

> > ### Author Response · Authors · 2019-11-12
> > **Answer to Reviewer 2 (2/2)**
> >
> > > 1. On page 3, the paper states “We are not the first to consider a similar learning problem, although most prior works do require some form of external supervision.” While there is a substantial body of literature that uses extra supervision, the following foundational works do not: Finn et al, 2016a,b; Fragkiadaki, 2016. In addition, “a similar learning problem” is considered without supervision in many papers on video prediction, including: Ranzato et al., 2014; Srivastava et al., 2015; Mathieu et al., 2015; Denton et al., 2017, 2018; Villegas et al., 2017, 2019; Wichers et al., 2018; Castrejon et al., 2019. While it might not be necessary to cite all of these as they do not necessarily apply their models to intuitive physics, the statement in the paper seems rather unsubstantiated.
> >
> > We already discussed in the related work (page 2) that both Finn et al 2016a and 2016b are unsupervised. We have updated the paper to clarify on Fragkiadaki’s work. While this work is also unsupervised, our task has an additional complexity of ball-obstacle interactions. Besides, this work does use implicitly a significant amount of supervision as they assume to know the generative model of the data (and this generative model is integrated in their solution). Instead, we also learn the generative model in an unsupervised manner. This therefore allow the objects to disappear and adds an additional complexity to the task. This is particularly challenging as maintaining an implicit representation of the object is generally failing over time. We have now clarified this significant difference in the related work.
> >
> > Finally, if video prediction papers mentioned by the reviewer are related to ours as they also try to predict future frames, they do not aim specifically to model physical interactions. This is a major difference with goal to make long term predictions on physical scenarios and learn physical interactions between objects.
> >
> > > 2. A similar meta-learning prediction problem is addressed in Nagabandi’19a,b, which are not cited.
> >
> > We thank the reviewer for this suggestion. We added it to the updated version of the paper in the related work. Note that these works are not concurrent to ours since they tackle online model adaptation which allows the model to learn and adapt at test time. Differently, in our work we model the environment from past observations only and uses a fixed, predicted, auto-regressor at test time.

---

### Official Review · AnonReviewer3 · 2019-10-22
**Official Blind Review #3**

**Rating:** 3

**Review:**

UPDATE: I appreciate the authors' discussions and qualitative results. My main original concern was that the empirical evaluation only studies a single type of situation of inferring physical parameters. Given that the authors claim that the proposed method infers "on the fly" physical properties, I would expect that the authors demonstrate the the method works on other physical properties as well beyond just one setting. Because this concern was not sufficiently addressed, I maintain my original rating.

----
Summary: This paper tackle the question of learning to infer physical parameters of a novel physical scenario with which to predict future motion of objects. In particular, the authors propose a meta-learning framework for inferring parameters of physical objects from a few video observations of physical scenario, with the constraint that physical parameters are not apparent from appearance alone. The authors evaluate the approach on a domain where the learner needs to infer whether balls should pass above, pass below, or bounce off obstacles.

Recommendation: Borderline. The main limitation of the paper is that the empirical evaluation only studies a single type of situation of inferring physical parameters (whether balls should pass above, pass below, or bounce off obstacles) but does not consider other situations for inferring physical parameters for prediction in a new domain. This paper has great potential, but the lack of evaluation on a wider variety of physical phenomena makes it difficult to evaluate the generality of the claims made by this paper. I would highly consider increasing my score if the authors would be able to provide a thorough evaluation of two more types of physical phenomena.

Research Problem: The research problem this paper tackles is to learn a model of intuitive physics that can learn "on the fly" physical properties specific to new environments.

Approach: A meta-network compresses a set of video observations of a scenario using the dynamic image encoding. The meta-network produces context parameters m and a that parameterize the scenario-specific predictor \hat{Phi} and conditional generator g respectively. The state x is represented with a two-dimensional array. The authors use a perceptual loss as an image reconstruction metric. The author also enforce that the predictions of the state-space predictor and the encoder eta should provide close predictions.

Strengths
- The meta-learning formalization is an interesting and intuitive contribution.
- The experiments provide a proof-of-concept of the efficacy of the framework with thorough analysis.

Weaknesses
- While the work is well-motivated and the experiments provided show a proof-of-concept of the approach with thorough analysis, the paper could be significantly strengthened by considering more domains for inferring physical parameters - simply inferring whether balls will pass above, pass below, or bounce off obstacles is a good first step, but does not provide enough evidence to evaluate the generality of the proposed method to intuitive physics tasks. This I believe is the main limitation of the paper.

Questions
- Would the authors be able to provide an empirical study (with qualitative analysis) or theoretical justification to explain the reason why the method can generalize to scenarios with more objects than observed during training? Prior work that does show generalization to more objects explicitly build in the locality constraint (e.g. van Steenkiste 2018) through the network architecture, so it is interesting to see that the proposed method also can generalize similarly.

Van Steenkiste, S., Chang, M., Greff, K., & Schmidhuber, J. (2018). Relational neural expectation maximization: Unsupervised discovery of objects and their interactions. arXiv preprint arXiv:1802.10353.

**Experience Assessment:**

I have published one or two papers in this area.

**Review Assessment: Checking Correctness Of Derivations And Theory:**

I assessed the sensibility of the derivations and theory.

**Review Assessment: Checking Correctness Of Experiments:**

I assessed the sensibility of the experiments.

**Review Assessment: Thoroughness In Paper Reading:**

I read the paper thoroughly.

---

> ### Author Response · Authors · 2019-11-12
> **Answer to Reviewer 3**
>
> We would like to thank the reviewer for comments and helpful suggestions.
>
> >- While the work is well-motivated and the experiments provided show a proof-of-concept of the approach with thorough analysis, the paper could be significantly strengthened by considering more domains for inferring physical parameters - simply inferring whether balls will pass above, pass below, or bounce off obstacles is a good first step, but does not provide enough evidence to evaluate the generality of the proposed method to intuitive physics tasks. This I believe is the main limitation of the paper.
>
> While we agree that applying our system to more tasks would be interesting, in practice, we believe the scenarios and problem tackled in this work were also studied in a more supervised form in the intuitive physics literature (Fragkiadaki et al 2016, Chang et al 2017). In either work, authors assumed either the rendering system to be known or directly regressed on the object position with known obstacle positions. In contrast, we are not restricted by any of these assumptions, making it a much more challenging setup.
>
> > Would the authors be able to provide an empirical study (with qualitative analysis) or theoretical justification to explain the reason why the method can generalize to scenarios with more objects than observed during training? Prior work that does show generalization to more objects explicitly build in the locality constraint (e.g. van Steenkiste 2018) through the network architecture, so it is interesting to see that the proposed method also can generalize similarly.
>
> Thanks for the suggestion, we already provide such an empirical study (table A3).
> Our work only used a propagation mechanism which used a shallow fully convolutional network to propagate the state of the system. This was done to keep the internal state spatially distributed. Therefore the receptive field of the network was not big enough to merge two objects together when they were further apart on the board. This was empirically verified in Table A3. When training with 1 ball and testing with 3 or 5 balls our module was able to propagate the objects until they were merged when physically close. When training with 3 balls and testing with 5 the number of detected objects overtime remain stable at around 5.

---

> > ### Comment · AnonReviewer3 · 2019-11-14
> > **Response**
> >
> > Thank you for your reply.
> >
> > > While we agree that applying our system to more tasks would be interesting, in practice, we believe the scenarios and problem tackled in this work were also studied in a more supervised form in the intuitive physics literature (Fragkiadaki et al 2016, Chang et al 2017). In either work, authors assumed either the rendering system to be known or directly regressed on the object position with known obstacle positions. In contrast, we are not restricted by any of these assumptions, making it a much more challenging setup.
> >
> > While it is true that the authors' setup is not restricted by assumptions about known obstacle positions, the comparison between regressing object positions and inferring physical properties seems to be a bit irrelevant to the discussion. Given that the authors claim that the proposed method infers "on the fly" physical properties, I would expect that the authors demonstrate the the method works on other physical properties as well beyond just one setting. For example, would the method be able to infer the unknown mass or magnetic properties of an object, both of which cannot be immediately inferred from perception? If the authors would be able to show experiments for these two settings, this would more strongly show the generality of the authors' claims. Otherwise, with only results on a single domain it is not clear whether the authors method learns "on the fly" physical properties in the general sense.
> >
> > Would it be possible to provide a link to the predicted videos (comparing to ground truth) corresponding to Table A3? As generalization to different numbers of objects relates the paper's claims about locality, it would be useful to qualitative assess how the proposed method generalizes. I ask because there is some work such as van Steenkiste et al, or Kosiorek et al that specifically build in locality constraints that enable generalizing to different numbers of balls. The paper suggests that a shallow fully convolutional network may be sufficient to achieve similar generalization, so it would be useful to have a baseline comparison with either van Steenkiste et al or Kosiorek et al.
> >
> > Kosiorek, A., Kim, H., Teh, Y. W., & Posner, I. (2018). Sequential attend, infer, repeat: Generative modelling of moving objects. In Advances in Neural Information Processing Systems (pp. 8606-8616).

---

> > > ### Author Response · Authors · 2019-11-15
> > > **Answer to request**
> > >
> > > We again, thank the reviewer for insightful comments and fruitful discussion.
> > >
> > > As requested we have updated the code link with an additional video of the multiple balls generalisation. The video contains sequences on random scenarios for 3 and 5 balls for the completely unsupervised network that was trained with only 3 balls.
> > >
> > > We note that even in the case of 5 balls our framework was still able to handle collisions between objects in general, although the network has never seen as many objects during training.

---

### Decision · Program_Chairs · 2019-12-19

**Decision:**

Reject

**Comment:**

While the reviewers found the paper interesting, all the reviewers raised concerns about the fairly simple experimental settings, which makes it hard to appreciate the strengths of the proposed method. During rebuttal phase, the reviewers still felt this weakness was not sufficiently addressed.